# Profiling of phytochemicals from aerial parts of *Terminalia neotaliala* using LC-ESI-MS$^2$ and determination of antioxidant and enzyme inhibition activities

**Muhammad Nadeem Shahzad**[1]*, **Saeed Ahmad**[1]*, **Muhammad Imran Tousif**[2], **Imtiaz Ahmad**[1,3], **Huma Rao**[1], **Bilal Ahmad**[1], **Abdul Basit**[1]

**1** Department of Pharmaceutical Chemistry, The Islamia University of Bahawalpur, Bahawalpur, Pakistan, **2** Department of Chemistry, D.G. Khan Campus University of Education, Lahore, Pakistan, **3** Department of Medicinal Chemistry, College of Pharmacy, University of Minnesota, Minneapolis, Minnesota, United States of America

* shazad_sca@yahoo.com (MNS); rsahmed_iub@yahoo.com (SA)

## Abstract

### Objectives

Owing to extraordinary healing power, *Terminalia* species have been used in traditional medicine systems to treat various diseases. Many folklore uses of *Terminalia neotaliala* (Madagascar's almond) included treating arterial hypertension, diabetes, diarrhea, dysentery, colic, oral and digestive candidiasis, intestinal parasitic infections, inflammatory skin conditions, postpartum care, and mycotic infections but nevertheless scientifically explored for its medicinal and pharmacological importance. Therefore, the current study intended to prepare methanolic extract and its fractionation with hexane, chloroform, and butanol followed by evaluation of their polyphenolic content, biological activities, and LCMS analysis. The biological study included antioxidant activity and enzyme inhibition assay i.e., α-glucosidase and urease. The insight study of biologically active secondary metabolites of butanol fraction (BUAE) was performed through LCMS.

### Methods

The total phenolic content (TPC) and total flavonoid content (TFC) of hydroalcoholic and its fractions were estimated using the Folin-Ciocalteu and aluminum chloride method. The total tannin content (TTC) was determined using the Folin-Denis spectrophotometric method. Similarly, the antioxidant potential of HAAE, HEAE, CFAE, and BUAE was determined using four methods as DPPH (1,1-diphenyl-2-picrylhydrazyl), 2,2-azinobis(3-ethylbenothiazoline)-6-sulfonic acid, cupric reducing antioxidant capacity (CUPRAC), and ferric reducing antioxidant power (FRAP). The sample extracts were also evaluated against two clinically important enzymes i.e., α-glucosidase and urease.

**Data Availability Statement:** All relevant data are within the manuscript.

**Funding:** The authors received no funding for this work.

**Competing interests:** The authors declare that no competing interests are involved in this study.

## Results

The BUAE (butanol aerial fraction) showed the highest TPC (234.79 ± 0.12 mg.GAE.g$^{-1}$ DE), TFC (320.75 ± 12.50 mg.QE.g$^{-1}$ DE), and TTC (143.36 ± 4.32 mg.TA.Eq.g$^{-1}$ DE). The BUAE also showed the highest scavenging potential determined by DPPH (642.65 ± 1.11 mg.TEq.g$^{-1}$ DE) and ABTS (543.17 ± 1.11 mg.TEq.g$^{-1}$ DE), and the metal-reducing capacity determined by CUPRAC (1510.41 ± 4.45 mg.TEq.g$^{-1}$ DE) and FRAP (739.81 ± 19.32 mg. TEq.g$^{-1}$ DE). The LCMS of BUAE identified 18 different biologically active phytoconstituents validating a rich source of hydrolyzable tannins including ellagitannins and gallitannins.

## Conclusion

The present study concluded that *T. neotaliala* is a rich source of polyphenols capable of neutralizing the damage caused by free radical accumulation in the cells and tissues. The significant antioxidant results and identification of high molecular weight hydrolyzable tannins enlightened the medicinal importance of *T. neotaliala*.

## 1 Introduction

In consonance with a WHO report, approximately 80 percent of the world's population relies on medicinal plants and herbal drugs [1]. The phytochemicals worked as defense soldiers to maintain human and animal health by regulating metabolic systems [2]. The ROS (reactive oxygen species) produced during metabolic processes negatively affected cellular components and disturbed their metabolic functions [3, 4]. The antioxidant prevented the accumulation of highly reactive oxygen species (ROS) and protected the cells against oxidative damage [5].

The production of scavengers, endogenous enzymes, and hormones [6] produced during the inflammatory disorders was found incapable to fight against excessive accumulation of highly ROS. The plants were reported rich sources of natural antioxidants which augmented the production of these endogenous molecules and protected the body from the negative effects of ROS [7].

Over the last three decades, more than 4000 antioxidants had been isolated from medicinal plants which include polyphenols such as flavonoids, proanthocyanidins, phenolic acids, tocopherols, tannins, coumarins, and anthraquinones [8, 9]. These phenolic compounds showed antioxidant activity owing to oxidation-reduction potential and metal-chelating effect [10].

*Terminalia* is the second-largest genus including 200 species [11]. The word "*Terminalia*" originated from the Latin word terminus (end or extremity of a thing, boundary, or limit) because the leaves grow at the tips of their shoots [12]. The *Terminalia neotaliala* (*T. mantaly* H. Perrier, *T. obscordiformis*) belongs to the family "Combretaceae" and is vernacularly known as Madagascar's almond or the umbrella tree. It is a large endemic tree that grows to 20 m height in dry, humid, and subhumid forests at elevations of 0–500 m [10].

An extensive literature survey revealed its ethnobotanical use for treating arterial hypertension, gastroenteritis, diarrhea, dysentery, diabetes, colic, oral and digestive candidiasis, intestinal parasitic infections, inflammatory skin conditions, postpartum care, and mycotic infection. According to another study, leaf, and bark extracts of *T. neotaliala* were also used against hypertension, gastroenteritis, diabetes, skin conditions, oral and genital candidiasis [13].

The *Terminalia* species have been screened for their antioxidant potential by DPPH, superoxide radical scavenging, and FRAP assays and showed high antioxidant potential [11]. But

the phytochemical and biological activity of *T. neotaliala* has not been detailed evaluated. In the present study, we prepared hydroalcoholic extract and its different solvent fractions followed by evaluation of their polyphenolic content, pharmacological activities such as antioxidant and enzyme inhibition. Qualitative metabolomic profiling of the butanol fraction of *T. neotaliala* was also performed using LCMS.

## 2 Materials and methods

### 2.1 Chemicals

Ascorbic acid, DPPH, quercetin, gallic acid, Folin–Ciocalteu reagent, ethanol, and methanol were procured from Merck (Darmstadt, Germany). Trolox (6-hydroxy-2, 5, 7, 8-tetramethyl-chroman-2-carboxylic acid), aluminum chloride, sodium nitrite, sodium hydroxide, sodium dihydrogen phosphate, neocuproine, $FeCl_2$, ABTS, ferrozine, EDTA, α-glucosidase, and urease were purchased from Sigma-Aldrich (St. Louis, MO, USA).

### 2.2 Plant material

The aerial parts of *T. neotaliala* were collected from different nurseries and gardens located in Bahawalpur, Punjab-Pakistan 63100, from September to October 2018. A taxonomist from the Herbarium Department of Botany-Faculty of Life Sciences, the Islamia University of Bahawalpur, identified the plant as *Terminalia neotaliala* and issued a voucher (59/botany dated the Bahawalpur 25[th] -Sep-2018).

### 2.3 Processing of plant material

The collected aerial parts were shade-dried and pulverized using a grinding mill. The powder was preserved in a hermetically sealed container and placed in a cool, dry, and dark area.

### 2.4 Preparation of plant extract

The 7 Kg dried powder was macerated in 80% hydroalcoholic solution for two weeks with occasional stirring. The supernatant was collected and filtered using Whatman filter paper no.1 followed by fine filtration through the Buchner funnel. The filtrate was concentrated and dried at 45˚C using a rotary evaporator under reduced pressure. The extract was suspended in water and fractionated by liquid-liquid extraction using hexane, chloroform, and butanol in increasing order of their polarity. The HEAE (hexane aerial fraction 20 g), 'CFAE' (chloroform aerial fraction 120 g), and 'BUAE' (butanol aerial fraction 90 g) sample extracts were secured through fractionation of HAAE (hydroalcoholic aerial extract 255g).

### 2.5 Phytochemical screening

The phytochemical analysis of the methanolic extract and its fractions was performed to identify primary and secondary metabolites. The screening for primary metabolites included carbohydrates, proteins, lipids, and amino acids, and screening for secondary metabolites included phenols, tannins, flavonoids, alkaloids, glycosides, steroids, saponins, terpenes, and resins.

### 2.6 Total phenolic content (TPC)

The TPC of sample extract was determined using the Folin-Ciocalteu reagent (FCR) method [14]. 1mL sample extract (1 mg.L$^{-1}$) was vigorously mixed with an equal volume of diluted FCR. After 5 min, 0.75 mL of 1% sodium carbonate solution was added to the mixture and incubated at ambient temperature for 2 h. The absorbance was recorded at 760 nm. The same

procedure was adopted for different aliquots of gallic acid. The results were expressed as milligram gallic acid equivalent per gram of dry extract (mg.GAEq.g$^{-1}$ DE).

## 2.7 Total flavonoid content (TFC)

The TFC of sample extract was determined using the aluminum chloride ($AlCl_3$) method [15]. 1 mL sample extract was mixed with an equal volume of 2% $AlCl_3$. The mixture was incubated at ambient temperature for 10 min and absorbance was recorded at 415 nm. The same procedure was adopted for the different aliquots of quercetin. The results were expressed as mg. QuEq.g$^{-1}$ DE.

## 2.8 Antioxidant activity

The antioxidant activity of each sample extract of *T. neotaliala* was determined using radical-scavenging activity and reducing power. The results were expressed as milligram Trolox equivalent per gram of dry extract (mg. TEq.g$^{-1}$ DE).

**2.8.1 Radical scavenging activity.** The radical scavenging potential was determined by DPPH and ABTS methods.

*2.8.1.1 DPPH scavenging activity*. The 50 μL of each sample extract was added to a 96 well plate followed by the addition of 150 μL 400 mM DPPH solution [14]. The mixture was kept in dark at ambient temperature for 30 min. The absorbance was recorded at 517 nm. A similar procedure was used for the different aliquot solutions of Trolox. The results were expressed as milligram Trolox equivalent per gram of dried extract (mg.TEq.g$^{-1}$ DE).

*2.8.1.2 ABTS scavenging activity*. The ABTS scavenging potential of the sample extract was determined by the formation of the ABTS (2, 2-azino-bis (3-ethylbenzothiazoline) 6-sulfonic acid) $^{+}$ radical cation [14] when a mixture of 7 mM ABTS and 2.45 mM potassium persulfate was kept in the dark at ambient temperature. ABTS$^{+}$ solution was diluted with methanol until the absorbance value was recoded 0.700 at 734 nm. The 1mL of sample extract solution was mixed with 2 mL ABTS$^{+}$ solution and incubated for 30 min followed by absorbance measurement at 734 nm. The results were expressed as milligram Trolox equivalent per gram of dried extract (mg. TEq.g$^{-1}$ DE).

**2.8.2 Reducing power assays.** The reducing power of sample extract was evaluated by two different methods.

*2.8.2.1 CUPRAC*. The CUPRAC reducing activity was determined using a previously reported method with some modifications [15]. The 3 mL reaction mixture [10 mM $CuCl_2$, 7.5 mM neocuprion, and 1 M ammonium acetate buffer (pH 7), 1:1:1] was mixed with 0.5mL sample solution (1 mg.mL$^{-1}$) and incubated at room temperature for 30 min. The absorbance of the resulting solution was recorded at 450 nm. The results were expressed as milligram Trolox equivalent per gram of dry extract (mg. TEq.g$^{-1}$ DE).

*2.8.2.2 FRAP*. The FRAP reducing activity was determined using a previously described method [16] with minor modifications. 2 mL reaction mixture [0.3 M acetate buffer (pH 3.6), 20 mM $FeCl_3$, and 10 mM TPTZ solution in 40 mM HCl, 10:1:1] was mixed with 0.1 mL extract solution (1 mg.mL$^{-1}$). The resulting mixture was incubated at room temperature for 30 minutes. The absorbance was recorded at 593 nm. The results were expressed as milligram Trolox equivalent per gram of dry extract (mg. TEq.g$^{-1}$ DE).

## 2.9 Determination of total tannin content (TTC)

The total tannin content sample extract was determined using the Folin-Denis spectrophotometric method [17]. The 1g of sample extract was vigorously mixed with 10 mL distilled water and kept at room temperature for 30 min with occasional stirring. The mixture was

centrifuged, and the supernatant was collected. Then, 2.5 mL supernatant and 1 mL Folin-Denis's reagent were added in a 50mL volumetric flask followed by the addition of 2.5mL $Na_2CO_3$ saturated solution. The volume was made up to 50 mL and incubated at room temperature for 90 min. The absorbance was recorded at 250 nm using an IRMECO UV-Vis spectrophotometer (model U2020). The results were expressed as milligram tannic acid equivalent per gram of the dry extract (mg. $TAEq.g^{-1}$ DE).

## 2.10 In vitro enzyme inhibition potential

The in vitro biological activity of sample extract was evaluated against two clinically important enzymes. i.e., α-glucosidase and urease.

**2.10.1 α-glucosidase inhibition assay.** The α-glucosidase inhibition assay was performed according to a previously described method [17] with some modifications. The 10 μL enzyme solution ($1U.mL^{-1}$) and 50 μL of 50 mM phosphate buffer (pH 6.8) were added to the 96-well plate. 20 μL of the sample extract solution was added to the above mixture and incubated at ambient temperature for 15 min. The absorbance was recorded at 405 nm (pre-read). After that, 20 μL of 0.5 mM para-nitro-α-d-glucopyranoside (substrate) solution was added to the reaction mixture and again incubated at 37˚C for 30 min. The absorbance was recorded at 405 nm (after read), and the inhibition (%) was calculated using the following formula

$$\alpha\text{-glucosidase inhibition (\%)} = \frac{(1 - abs. \ of \ sample)}{abs. \ of \ control} \ x \ 100 \tag{1}$$

**2.10.2 Anti-urease activity.** The anti-urease activity of sample extract was determined using a previously reported method with some modifications [18]. The 20 μL of 0.025% urease in 1M phosphate buffer (pH 7.0) and 20 μL of sample solution was added to the same well of 96 microtiter plate and incubated at 37˚C for 15 min. Then, 60 μL of 2.25% urea solution was added to the above mixture and incubated again at 37˚C for 15 min. The absorbance was measured at 630 nm (pre-read). The 60 μL phenol reagent and 100 μL sodium hypochlorite solution (prepared in alkali) were added to the reaction mixture. The absorbance was recorded at 630 nm. The enzyme inhibition (%) was calculated using the following formula:

$$\textbf{Urease inhibition (\%)} = \frac{(\textbf{1} - \textbf{abs. of sample})}{\textbf{abs. of control}} \ x \ 100 \tag{2}$$

## 2.11 Phytochemical analysis of *T. neotaliala* by LC-ESI-ITMS$^2$

The insight of bioactive metabolites of BUAE was performed by using tandem mass spectrometry using an LTQ XL linear ion trap mass spectrophotometer (Thermo Scientific, USA) with an electron spray ionization (ESI) interface. The sample was prepared by dissolving the butanol fraction in methanol (MS grade). The Xcalibur data acquisition and interpretation software was used to operate the system. The system was equipped with a column (250 × 2.0 mm ODS-VP C18, 5 μm). Liquid chromatographic conditions included a mobile phase solvent A (0.1% formic acid) and solvent B (methanol). Gradient mode of elution was selected, starting with 5% B (0–5 min), 5–45% B (5–90 min), 45% B (90–100 min), 45–5% B (100–101 min).

The sample was filtered and injected using a direct syringe pump at a flow rate of 5 μL.min$^{-1}$. The scanning was performed in the positive and negative total ion full-scan modes in the mass scan range of m/z 50–2000. 4.8 kV source voltage and a 23 V capillary voltage were used during the process. Sheath glass flow (N2) was 30 arbitrary units, and the capillary temperature

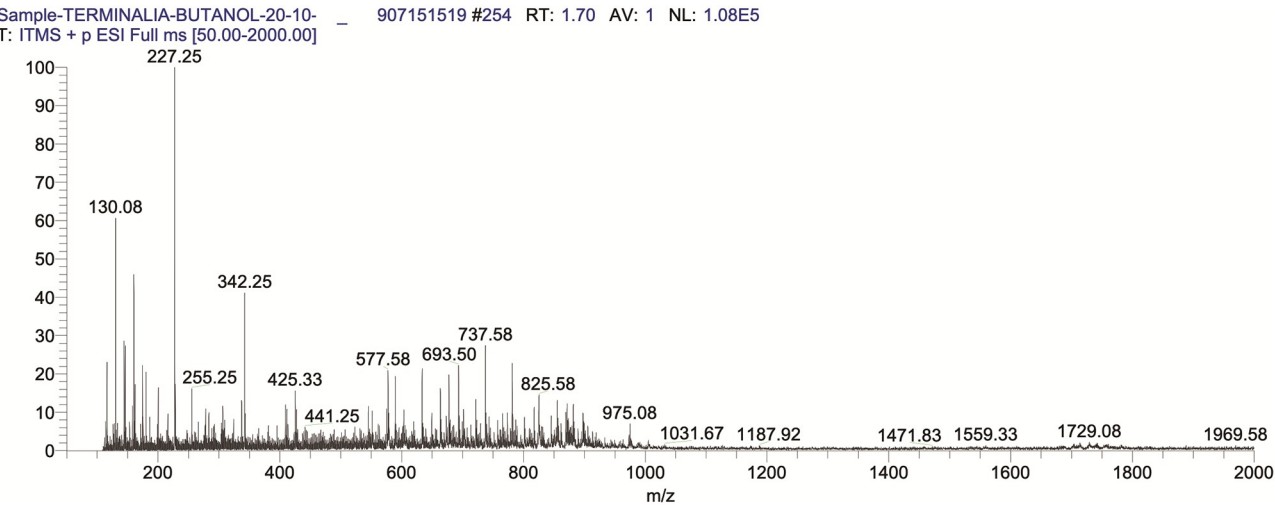

**Fig 1. TIC of BUAE in positive ion scan mode.**

was kept at 350˚C in positive and negative scan modes. For the analysis, fragmentation of the eluted compound was performed in both positive (Fig 1) and negative (Fig 2) ion mode using collision-induced dissociation energy of 35% (% of 5 V) [19, 20].

## 2.12 Statistical analysis

The statistical analysis was performed using IBM SPSS Statistics version 20. Each experiment was performed in triplicates and the result was expressed as a mean of triplicate values ± standard deviation (SD). The one-way analysis of variance (ANOVA) was applied to determine the significant difference between the results. The comparison among mean values was made using the least significant difference (LSD) to test for significant differences at $P < 0.05$.

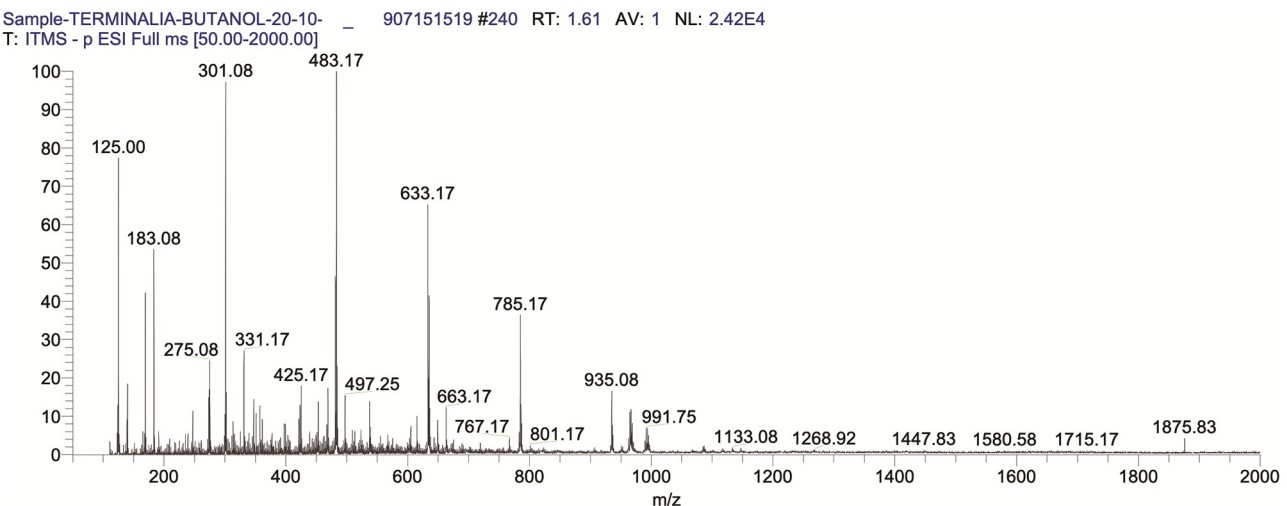

**Fig 2. TIC of BUAE in negative ion scan mode.**

**Table 1. Phytochemical screening of *T. neotaliala*.**

| Phytochemical Constituent | Name of Test | HAAE | HEAE | CFAE | BUAE |
|---|---|---|---|---|---|
| Carbohydrates | Molisch's test | +++ | +++ | +++ | +++ |
| Starch | Iodine test | +++ | +++ | +++ | +++ |
| Proteins | Biuret | +++ | +++ | +++ | +++ |
| Amino acids | Ninhydrin | +++ | +++ | +++ | +++ |
| Phenols | Lead acetate | +++ | +++ | +++ | ++++ |
| Alkaloids | Dragendorff's test | - | - | - | - |
| | Wagner test | - | - | - | - |
| | Mayer's test | - | - | - | - |
| | Erdmann's test | - | - | - | - |
| Flavonoids | Amyl alcohol | +++ | +++ | - | ++++ |
| Saponins | Frothing | - | - | - | - |
| Steroids and Terpenes | Salkowski's test | - | - | ++++ | ++++ |
| Glycosides | Grignard reaction | - | - | - | - |
| | Borntrager's test | - | - | - | - |
| Tannins | Ferric chloride | ++++ | ++++ | ++++ | ++++ |
| Resins | Acetic acid test | - | - | - | - |

(-) means absent, (+) means present

# 3 Results

## 3.1 Phytochemical analysis

The phytochemical analysis of sample extracts revealed the presence of primary and secondary bioactive secondary metabolites (Table 1).

## 3.2 Estimation of total phenolic content (TPC)

**3.2.1 Total phenolic content (TPC).** The TPC values were determined and presented in Table 2. The results were found between 6.85 ± 0.22 to 234.79 ± 0.12 mg. GA.Eq.g$^{-1}$ DE. The BUAE showed the highest TPC (234.79 ± 0.12 mg. GA.Eq.g$^{-1}$ DE) and CFAE showed the lowest TPC (6.85 ± 0.22 mg. GA.Eq.g$^{-1}$ DE). Statistically, it was revealed that each solvent used had a different capability to extract different phytoconstituents (p < 0.05).

## 3.3 Estimation of total flavonoid content (TFC)

The BUAE showed highest TFC 320.75 ± 12.50 mg. Qu.Eq.g$^{-1}$ DE) which was followed by HAAE 224. 92 ± 9.46 mg. Qu.Eq.g$^{-1}$ DE, HEAE 151.58 ± 10.10 mg. Qu.Eq.g$^{-1}$ DE, and CFAE 124.08 ± 6.29 mg. Qu.Eq.g$^{-1}$ DE (Table 2).

**Table 2.**

TPC, TFC, and TTC of sample extracts of *T. neotaliala*\* GA.Eq. stands for gallic acid equivalent; Qu.Eq. stands for quercetin equivalent; TA.Eq. stands for tannic acid equivalent; DE stands for dry extract. The expressed values were mean of triplicates ± SD. The mean values in the same column with different letters are significantly different (P < 0.05).

### 3.4 Estimation of total tannin content (TTC)

Similarly, The BUAE showed highest TTC 143.36± 4.32 mg. TA.Eq.$g^{-1}$ DE followed by CFAE 104.32 ± 4.22 mg.TA.Eq.$g^{-1}$ DE, HAAE 34.31b ± 1.42 mg. TA.Eq.$g^{-1}$ DE, and HEAE 17.54 ± 2.54 mg. TA.Eq.$g^{-1}$ DE.

### 3.5 Antioxidant activity

The antioxidant results revealed that BUAE > HAAE > HEAE > CFAE as determined by all four antioxidant assays. The results were statistically analyzed which revealed that the antioxidant content of each fraction is significantly ($p < 0.05$) different from others when determined by a single method. A significant relationship was found between the total phenolic content and the antioxidant activity concluding.

### 3.6 α -glucosidase and anti-urease inhibition assay

The α-glucosidase and urease inhibitory potential of sample extract was determined and their $IC_{50}$ values were also calculated (Table 3).

The BUAE exhibited higher α-glucosidase inhibition activity with $IC_{50}$ value 0.21± 0 mg. $mL^{-1}$ followed by HEAE $IC_{50}$ 0.56 ± 0.03 mg.$mL^{-1}$ and HAAE $IC_{50}$ 0.73 ± 0.12 mg.$mL^{-1}$. The HAAE showed lowest $IC_{50}$ 1.79 ± 0.14 mg.$mL^{-1}$ as compared to the BUAE $IC_{50}$ 2.18 ± 0.16 mg.$mL^{-1}$ and HEAE $IC_{50}$ 3.54 ± 0.12 mg.$mL^{-1}$.

### 3.7 LC-ESI-$MS^2$ analysis of *T. neotaliala*

The phytochemical insight of secondary metabolites of BUAE by LCMS revealed the presence of 18 different biologically active phytochemicals (Table 4).

## 4 Discussion

Owing to the healing power and antioxidant potential, *T. chebula* was known as the "king of medicine" in Tibet which has been a preferred choice in Ayurvedic Materia Medica [11]. The antioxidant activity of *T. arjuna* bark was determined by DPPH, superoxide radical quenching, and lipid peroxidation assays and the results revealed that the antioxidant activity was equal to the antioxidant activity of ascorbic acid [26].

The BUAE was screened for its biologically active secondary metabolites by using liquid chromatography-mass spectrometry (LCMS) and identifying eighteen different phytoconstituents. And most of these identified compounds belonged to the subclass of polyphenols known

**Table 3. α-glucosidase and urease inhibition assays of four solvent extracts of *T. neotaliala*.**

| Extract/Fractions | α-glucosidase | Urease |
|---|---|---|
| | $IC_{50}$ (mg.$mL^{-1}$) ± SD | $IC_{50}$ (mg.$mL^{-1}$) ± SD |
| HAAE | 0.73[c] ± 0.12 | 2.18[b] ± 0.16 |
| HEAE | 0.55[b] ± 0.03 | 3.54[c] ± 0.12 |
| CFAE | - | - |
| BUAE | 0.21[a] ± 0.09 | 1.79[a] ± 0.14 |
| Quercetin | 0.01 ± 0.11 | - |
| Hydroxyurea | - | 0.98 ± 0.12 |

The values expressed were mean of triplicates ± standard deviation; superscripts [a], [b] and [c] represent the values with significant difference ($p < 0.05$).

**Table 4. The tentative identification of compounds by LCMS analysis of BUAE.**

| No. | RT | Compound class | Formula | MW | [M-H]⁻ | [M+H]⁺ | Fragmentation pattern | Tentatively Identified compound |
|---|---|---|---|---|---|---|---|---|
| 1 | 2.93 | Di carboxylic acid | $C_{12}H_{22}O_4$ | 146.16 | - | 147.1 | 41,55,73,87,100,128 | Adipic acid, hexane di-oic-acid [21] |
| 2 | 17.76 | Phenolic acid | $C_7H_6O_5$ | 170.12 | 169.50 | - | 125, 169 | Gallic acid [22] |
| 3 | 17.94 | Gallate ester (Hydrolysable tannin) | $C_8H_8O_5$ | 184.15 | 183.50 | - | 124,168 | Methyl gallate [23, 24] |
| 4 | 18.21 | Poly cyclic | $C_7H_{12}O_6$ | 192.17 | 191.50 | - | 173,149,111,93 | Quinic acid [24–26] |
| 5 | 18.21 | Carboxylic acid | $C_6H_8O_7$ | 192.12 | 191.50 | - | 173,111 | Citric acid [22, 24, 26] |
| 6 | 18.21 | Phenolic Coumarin | $C_{10}H_8O_4$ | 192 | 191.50 | - | 176,191,148,120 | Scopoletin [27] |
| 7 | 19.27 | Flavonol | $C_{15}H_9O_7$ | 302.235 | 301.50 | - | 301,178, 179,173,151,107,93 | Quercetin [28, 29] |
| 8 | 19.27 | Phenolic acid | $C_{14}H_6O_8$ | 302 | 301 | - | 301,257,229,185 | Ellagic acid [25, 30] |
| 9 | 19.58 | Gallotannin (hydrolyzable tannin) | $C_{13}H_{16}O_{10}$ | 332.26 | 331.50 | - | 169,125,271,331 | Galloyl hexoside [24] |
| 10 | 20.15 | Hydrolysable tannin | $C_{21}H_{10}O_{13}$ | 470.28 | 469.50 | - | 425,301,271 | Valoneic-di lactone [30, 31] |
| 11 | 20.68 | Hydrolysable tannin | $C_{20}H_{20}H_{14}$ | 484 | 483.50 | - | 331,211,271,169 | Di-O-galloyl glucose [24] |
| 12 | 21.52 | Hydrolysable tannin | $C_{27}H_{22}O_{18}$ | 634 | 633.50 | - | 275,301,481, | Galloyl HHDP hexoside [24] |
| 13 | 21.52 | Ellagitannin (Hydrolysable tannin) | $C_{27}H_{22}O_{18}$ | 634 | 633 | | 463,301 | Corilagin (1-O-galloyl-3,6-O-HHDP-β-ᴅ-Glc) [24] |
| 14 | 22.52 | Phenyl propanoid (Hydroxy cinnamate) | $C_{36}H_{36}O_{13}$ | 676 | 675.50 | - | 588,513,675 | Feruloyl-O-p coumaroyl-O caffeoyl shikimic acid [26] |
| 15 | 22.75 | Hydrolysable tannin | $C_{34}H_{26}O_{22}$ | 786.56 | 785.5 | - | 633,419,301,275 | Tellimagrandin, PedunculaginII, (Digalloyl HHDP-hex) [22, 32] |
| 16 | 23.12 | Hydrolysable tannin | $C_{34}H_{26}O_{23}$ | 802 | 801 | - | 649,301,275 | Digalloyl-HHDP-glucuronide [33] |
| 17 | 23.44 | Hydrolysable tannin | $C_{41}H_{28}O_{26}$ | 936.64 | 935.50 | - | 917,783,633,571, 481,419,329,301 | Galloyl-bis HHDP hexoside [33] |
| 18 | 23.44 | Ellagitannin (Hydrolysable tannin) | $C_{41}H_{28}O_{26}$ | 936.64 | 935.50 | - | 633,615,659,571, 481 | Casuarinin [31, 32] |

as hydrolyzable tannins which included galloyl hexoside (Fig 3), valoneic-di-lactone (Fig 4), galloyl-HHDP-hexoside (Fig 5), tellimagrandin or Pedunculagin II (Digalloyl-HHDP-hex) (Fig 6), galloyl-bis-HHDP-hexoside (Fig 7), Di-O-galloyl-glucose (Fig 8), digalloyl-HHDP-glucuronide (Fig 9), galloyl tannin, casuarinin (Fig 10), corilagin (Fig 11). The flavanol included

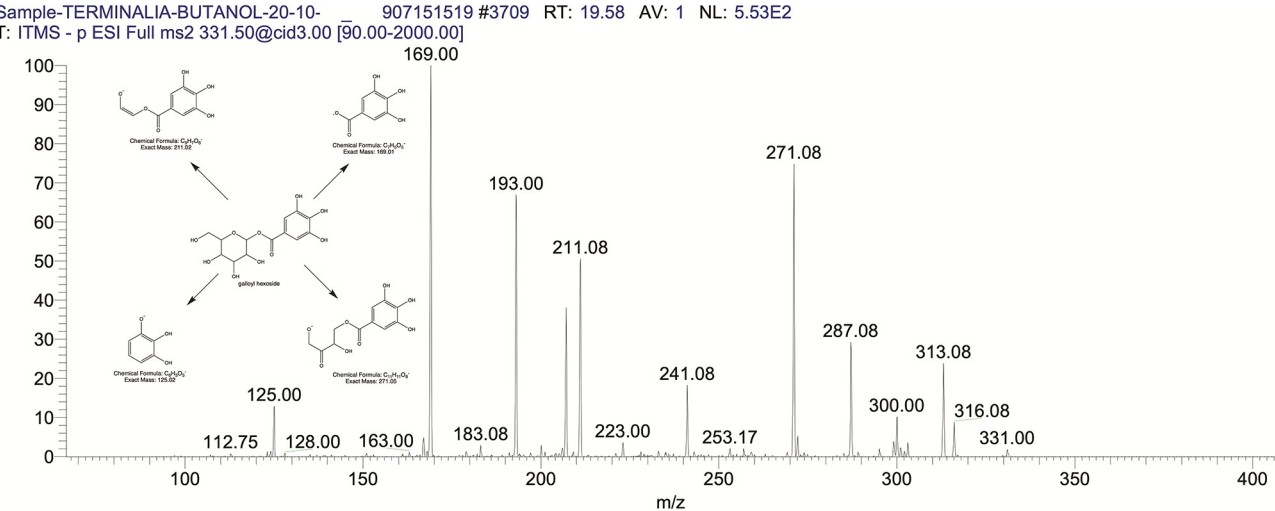

**Fig 3. The mass spectrum of galloyl hexoside and its fragmentation pattern.**

quercetin (Fig 12), phenolic acids included gallic acid (Fig 13) and ellagic acid (Fig 14), one methyl ester of gallic acid included methyl gallate (Fig 15), one carboxylic acid included citric acid (Fig 16), one dicarboxylic acid included adipic acid, hexane-di-oic-acid (Fig 17), one phenylpropanoid or hydroxy cinnamate included feruloyl-O-p-coumaroyl-O-caffeoyl shikimic acid (Fig 18), one cyclic polyol included quinic acid (Fig 19), and one coumarin included scopoletin (Fig 20).

The BUAE showed a significant amount of TPC (234.79 ± 0.12 mg. GA.Eq.g$^{-1}$ DE) and TFC (320.75 ± 12.50 mg. Qu.Eq.g$^{-1}$ DE) and TTC which concluded that *T. neotaliala* was a rich source of polyphenols as previously reported for other species of the genus *Terminalia*

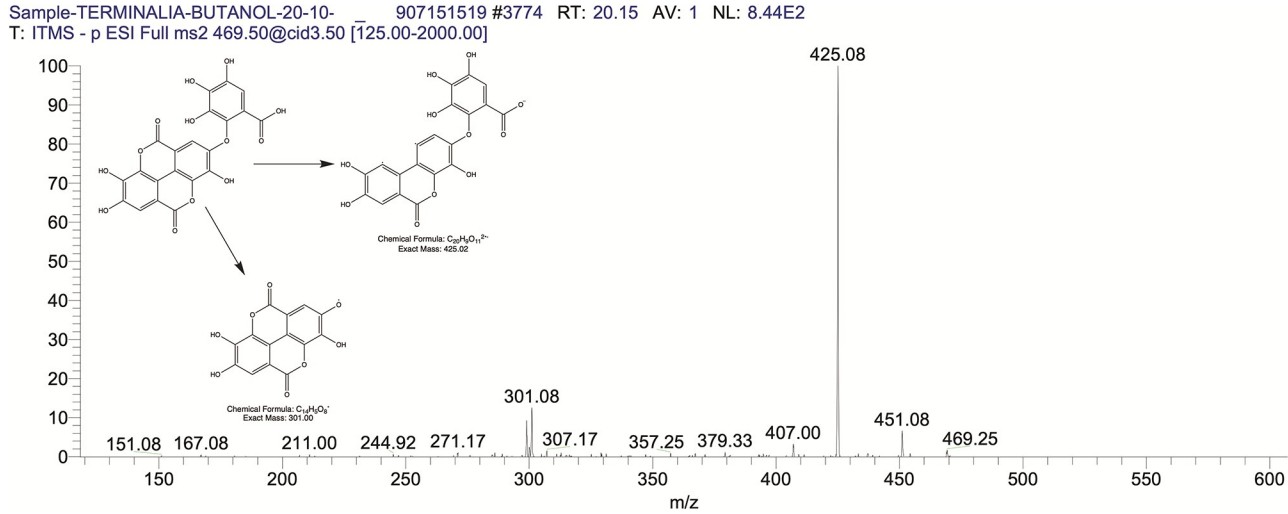

**Fig 4. The mass spectrum of valoneic- dilactone and its fragmentation pattern.**

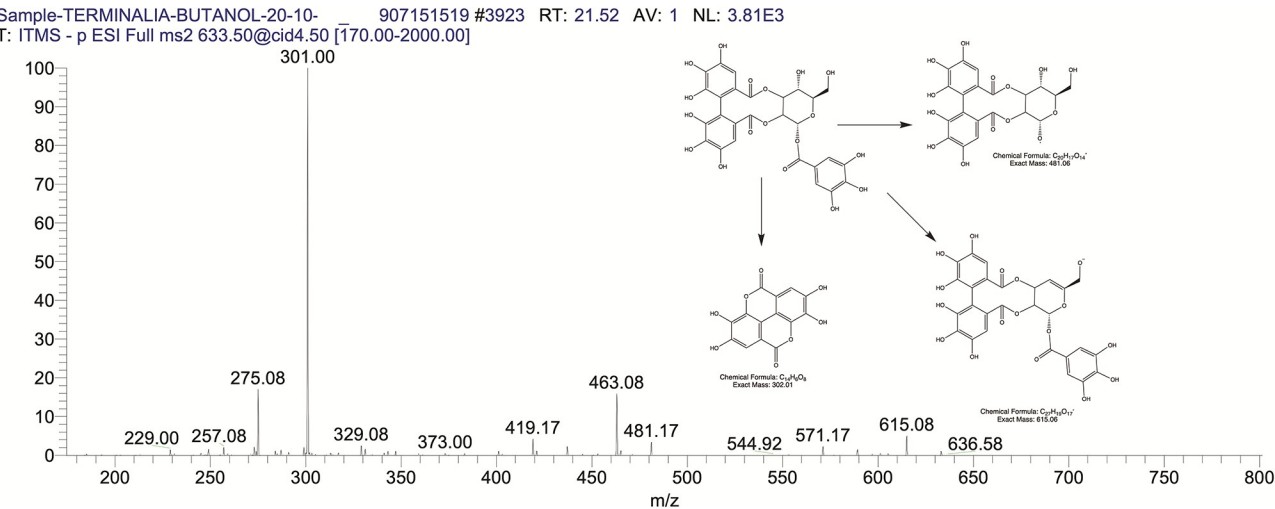

**Fig 5. The mass spectrum of galloyl HHDP hexoside and its fragmentation pattern.**

[25, 33]. Similarly, BUAE also showed strong antioxidant activity followed by HAAE, HEAE, and CFAE determined by all the described methods (Fig 21). *Terminalia* showed antioxidant potential owing to a wide variety of polyphenols and flavonoids [34, 35]. Polyphenol structures are ideal for free radical scavenging potential [36]. Tannins were abundant antioxidants in the human diet nevertheless they were often a neglected class of polyphenols. The TTC determination revealed that the highest level of tannins was present in the butanol fraction of the aerial parts of *T. neotaliala* (143.32 ± 4.32 mg. TA.Eq.g$^{-1}$ DE). The literature survey confirmed the presence of hydrolyzable and condensed tannins in genus *Terminalia* those who were reported with a high amount of TPC and TFC. The statistical analysis of TPC, TFC, TTC, and

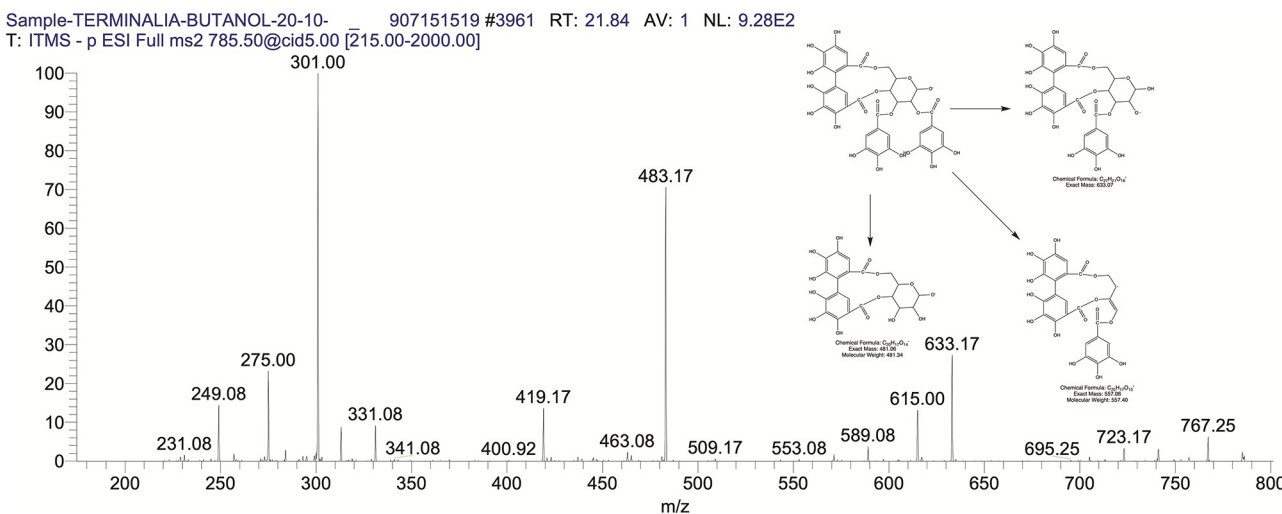

**Fig 6. The mass spectrum of tellimagrandin, pedunculaginII, (Digalloyl HHDP-hex) and its fragmentation pattern.**

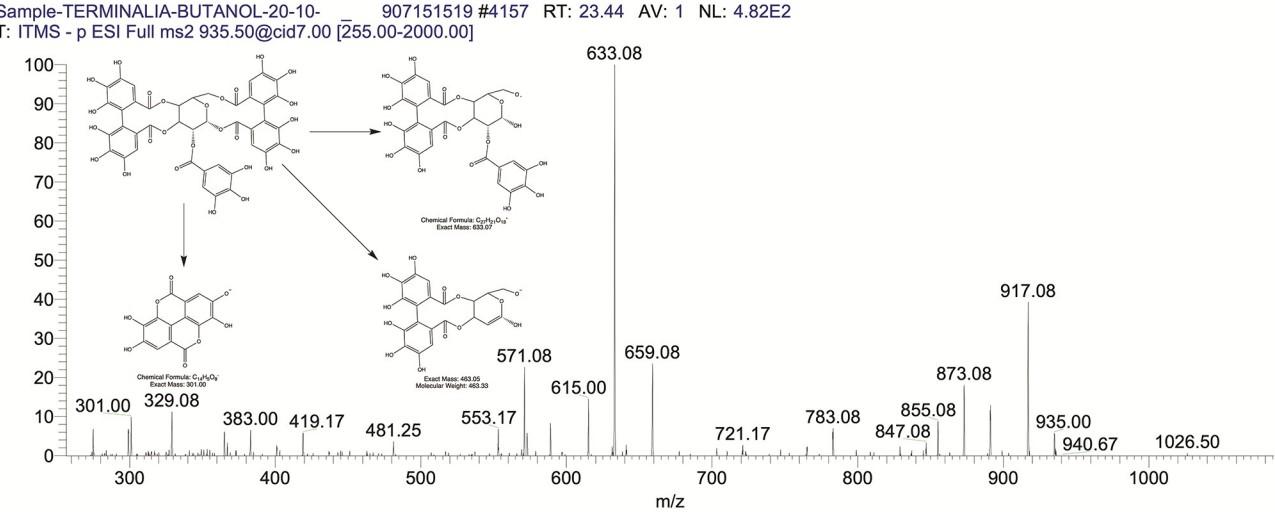

**Fig 7. The mass spectrum of galloyl-bis HHDP hexoside and its fragmentation pattern.**

antioxidant activity revealed that each sample extract solution i.e., HAAE, HEAE, CFAE, and BUAE; was significantly different from the other sample extract solution ($p < 0.05$). The tannins can function as chain-terminating antioxidants, metal chelators, reducing agents, scavengers of reactive oxygen species, and quenchers of singlet oxygen. The chloroform showed poor affinity to extract hydrolyzable tannins during extraction; therefore, CFAE exhibited the lowest polyphenolic content $6.86 \pm 0.22$ mg.GAE.g$^{-1}$ DE. The tannins, especially hydrolyzable tannins (HTs), constitute a major class of plant secondary metabolites, including simple gallic acid derivatives, gallotannins, and ellagitannins [37]. Tannins, especially hydrolyzable tannins, have attracted attention in recent decades because of their antioxidant potential and biological

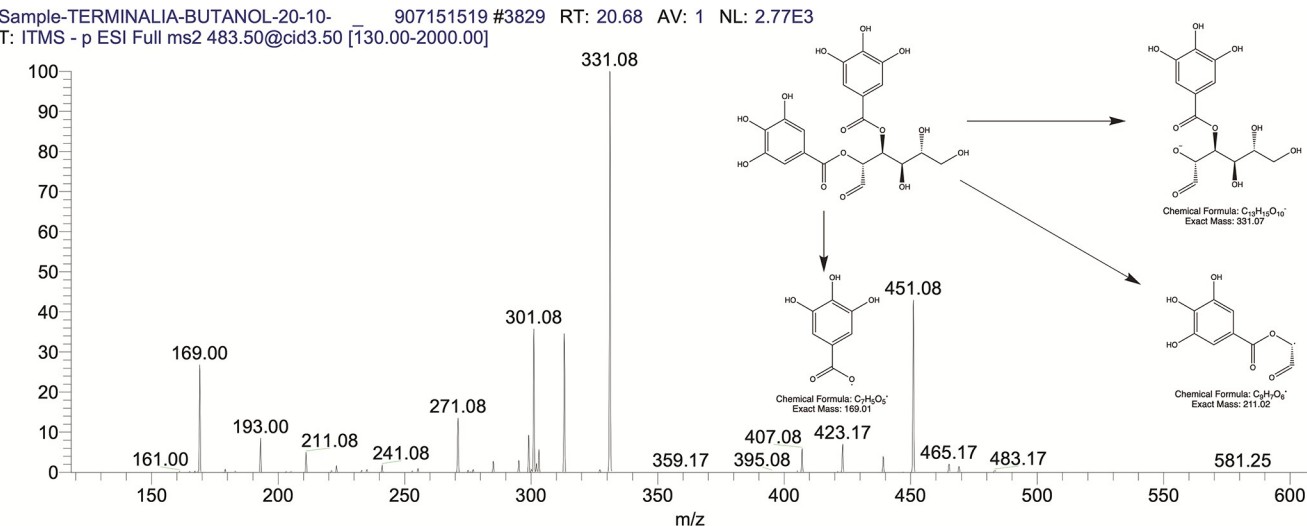

**Fig 8. The mass spectrum of di-O-galloyl glucose and its fragmentation pattern.**

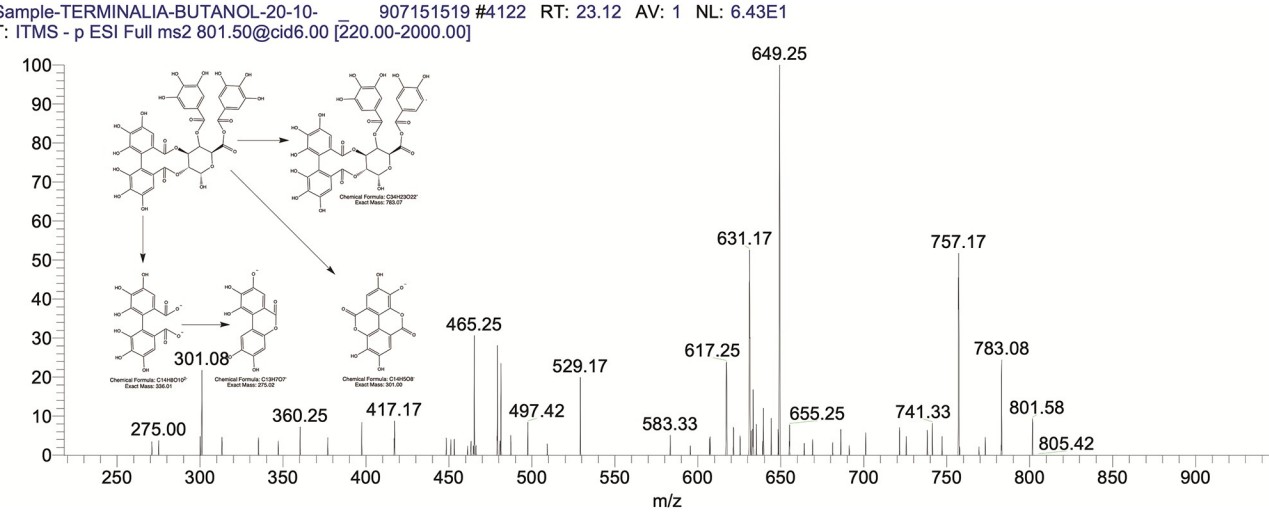

**Fig 9. The mass spectrum of digalloyl-HHDP-glucuronide and its fragmentation pattern.**

activity [38]. Many ellagitannins and hydrolyzable tannins, including ellagic acid, gallic acid, methyl gallate, ethyl gallate, chebulinic acid, corilagin, penta galloyl glucose, tetra galloyl beta-D-glucose, and casuarinin from the dichloromethane fraction of *T. chebula* fruit, improved type II collagen-induced arthritis in BALB/c mice [39]. These tannins were involved in the wound healing process by increasing the rate of epithelization [40].

Type II diabetes is one of the most prevalent metabolic disorders both in low and high-income countries being managed by improving insulin secretion and controlling blood glucose levels by administering oral hypoglycemic drugs such as α-glucosidase inhibitors, sulfonylureas, and biguanides. The inhibition of α-glucosidase found in the epithelium of the small

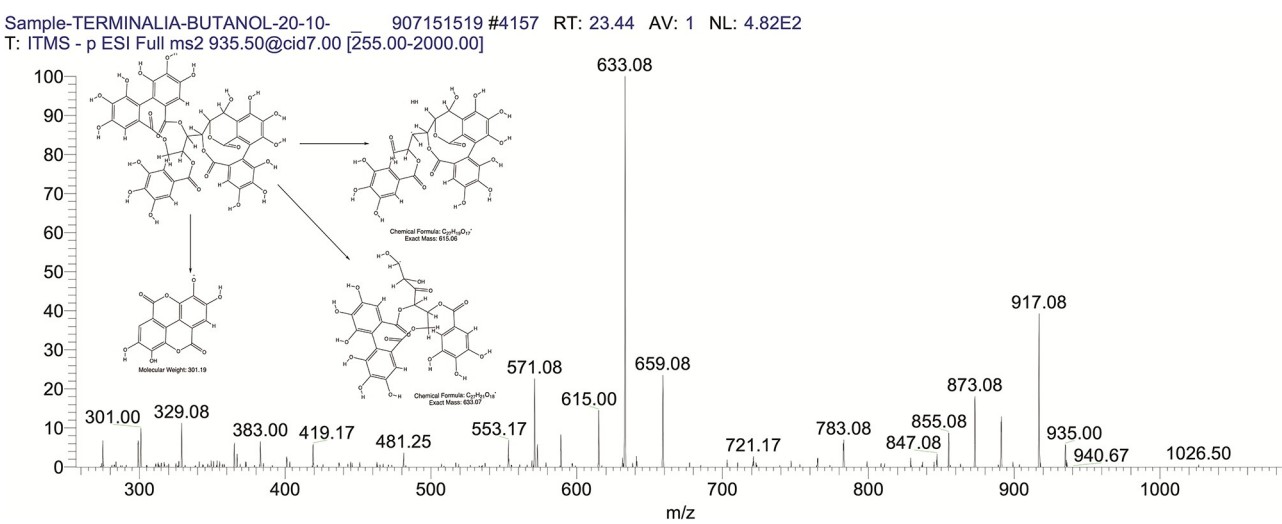

**Fig 10. The mass spectrum of casuarinin and its fragmentation pattern.**

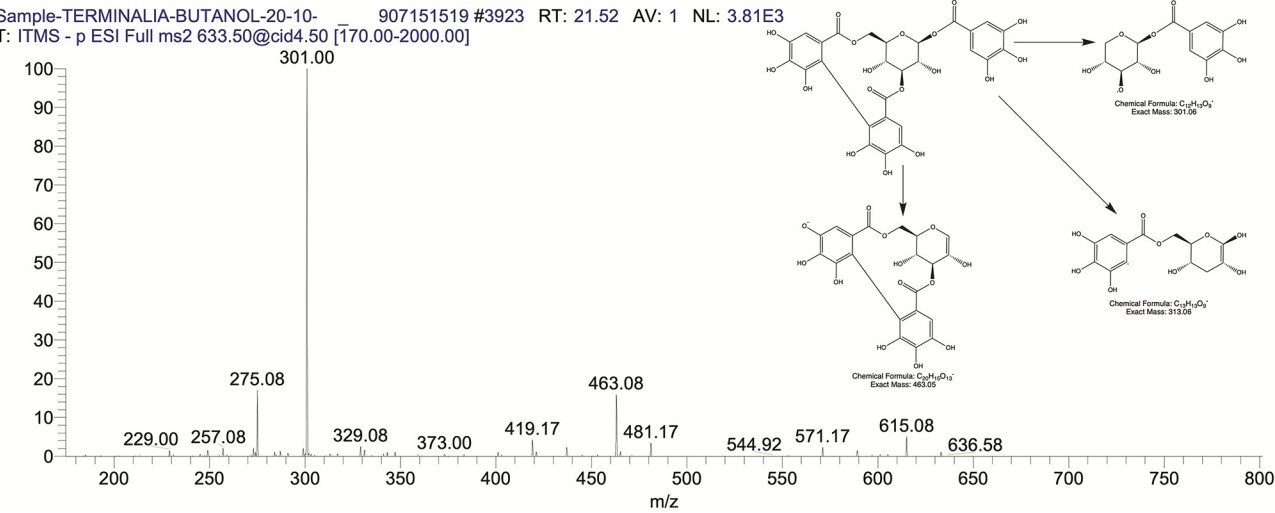

**Fig 11. The mass spectrum of corilagin (1-O-galloyl- 3,6-O-HHDP- β-D-Glc) and its fragmentation pattern.**

intestine delays the postprandial hyperglycemia by blocking the conversion of disaccharides into monosaccharides. There are only two marketed available α-glucosidase inhibitors i.e., acarbose and miglitol and therefore, there is an open field for the discovery of NMEs exploiting the natural resources. The BUAE significantly inhibited α-glucosidase with the lowest $IC_{50}$ value $0.21 \pm 0.09$ mg.mL$^{-1}$, followed by HEAE and HAAE. It was found through literature that the antidiabetic effects of the sample extract were owing to the ability of hydrolyzable tannins to block glucose intestinal epithelial uptake, glycogen, and lipid synthesis [41].

The urease is a virulence factor in human and animal pathogens that can cause kidney stones, pyelonephritis, peptic ulcers, urolithiasis, hepatic encephalopathy, and hepatic coma

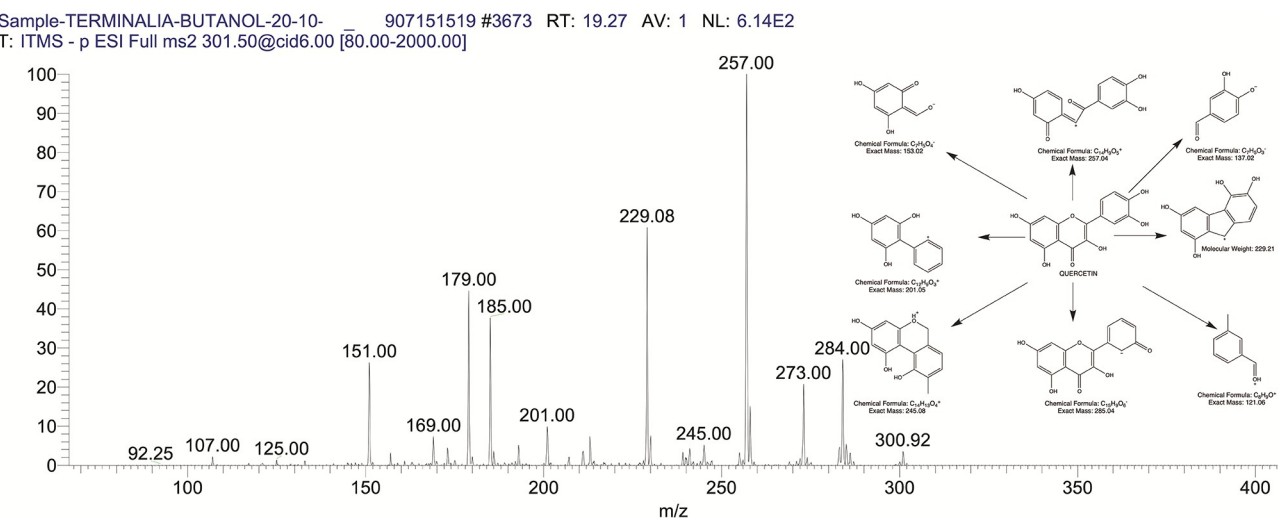

**Fig 12. The mass spectrum of quercetin and its fragmentation pattern.**

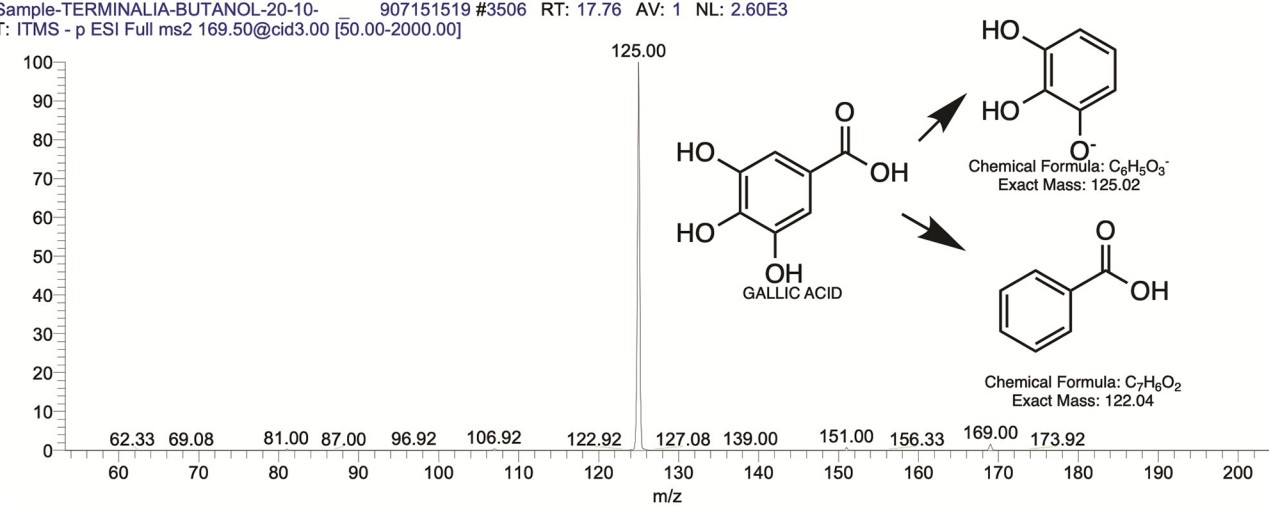

**Fig 13. The mass spectrum of gallic acid and its fragmentation pattern.**

[42]. The urease inhibitor prevents microorganisms from catalyzing urea into ammonia and carbon dioxide creating unfavorable conditions for their survival by restricting their major source of energy and nitrogen. This anti urease potential creates a good antimicrobial therapeutic option. The urease inhibitors have recently attracted attention owing to their anti-ulcerative potential [43]. Moreover, the recent increase in microbial antibiotic resistance and poor patient compliance necessitates the discovery of new inhibitors with better effectiveness and simpler regimens[44]. The anti-urease inhibitory potential was observed as BUAE > HAAE > HEAE> CFAE. The urease inhibitory activity of the BUAE may be attributed to the presence of hydrolyzable tannins and phenolic acids. The anti-urease activity of hydrolyzable tannin

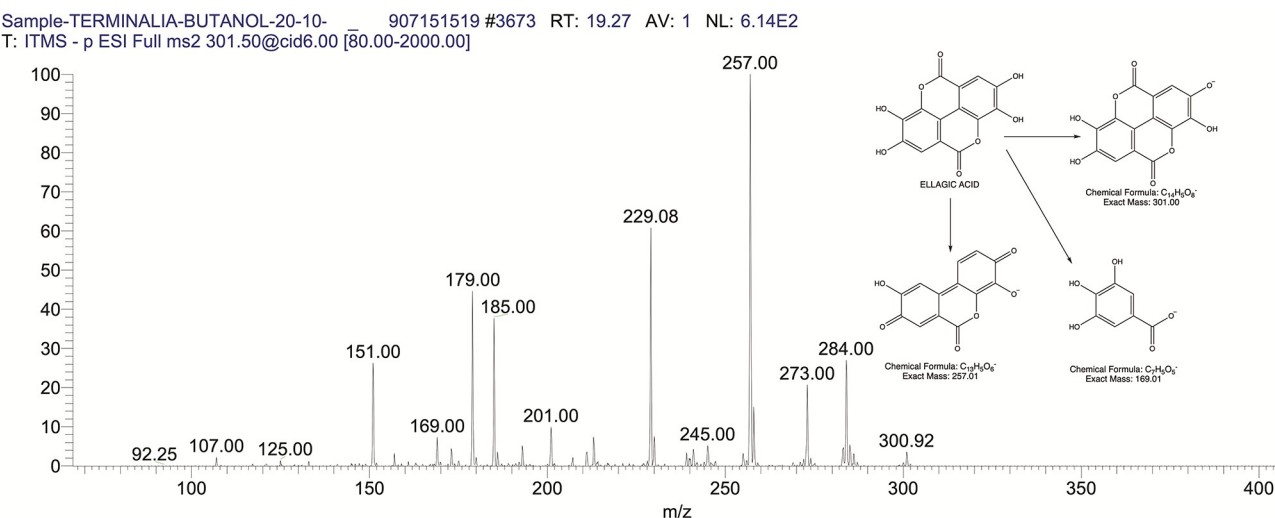

**Fig 14. The mass spectrum of ellagic acid and its fragmentation pattern.**

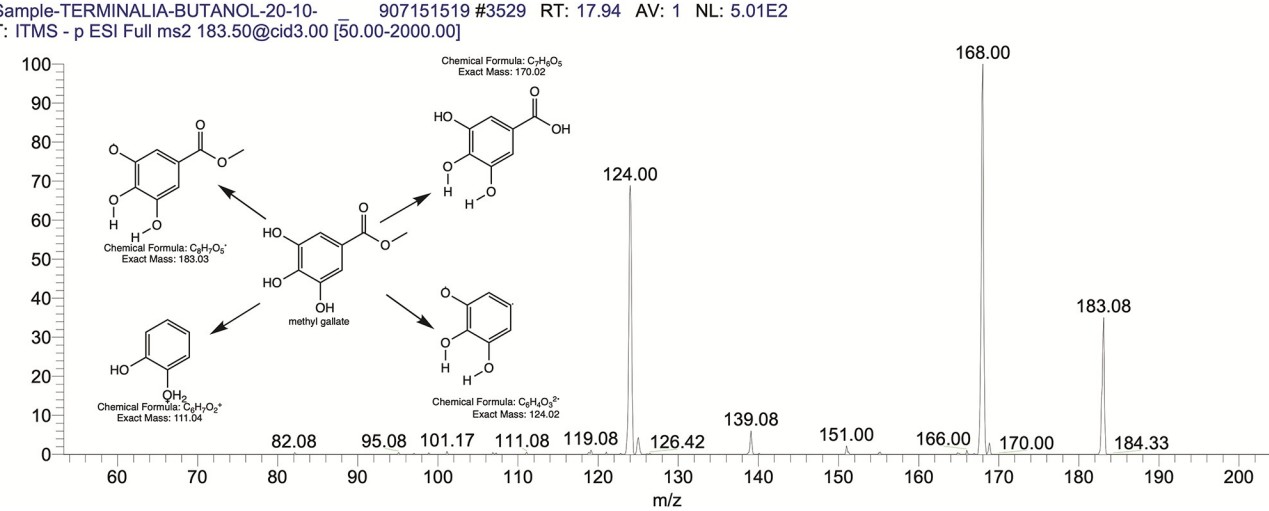

**Fig 15. The mass spectrum of methyl gallate and its fragmentation pattern.**

was due to the hydrogen bonding and hydrophobic interactions with the active site of urease [45]. A well-known flavanol i.e., quercetin, showed significant antioxidant, anticancer, anti-inflammatory, and antiviral activities. It has been considered therapeutically for its gastro-protective effects, inhibition of carcinogenicity, and reduction in the risk of cataracts. The allergic and anti-inflammatory response of quercetin was mediated by suppressing leukotriene production [46]. The corilagin inhibited glucose-6-phosphatase, fructose 1,6-bisphosphatase, and α-glucosidase [47]. These natural antioxidants may be promising drug candidates against various allergic and inflammatory conditions, metabolic disorders, and cancers. The quinic acid showed anticarcinogenic properties by up-regulating the cellular antioxidant enzymes and

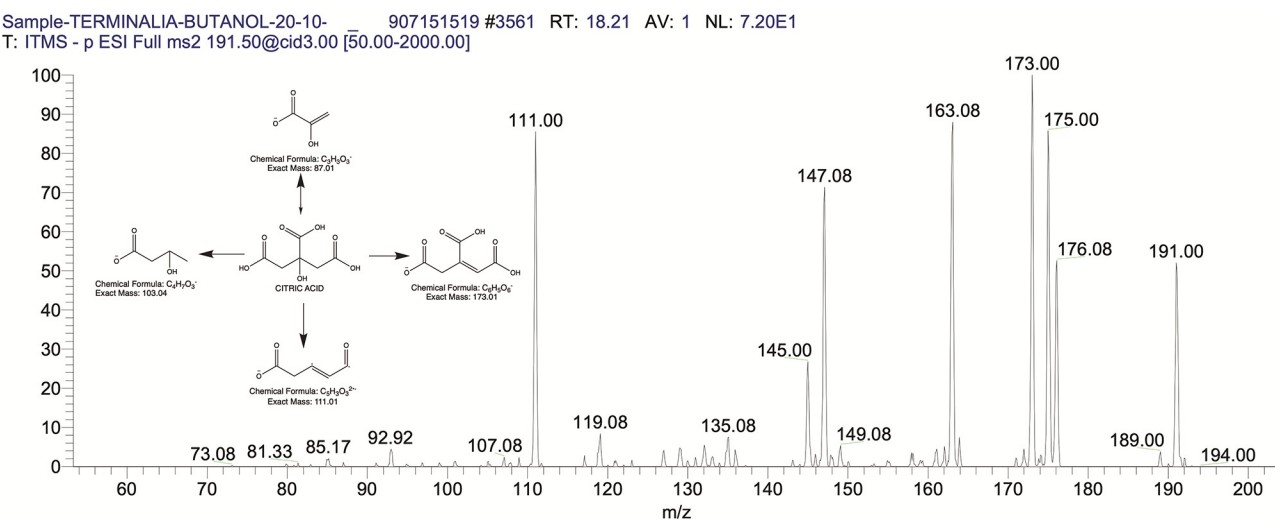

**Fig 16. The mass spectrum of citric acid and its fragmentation pattern.**

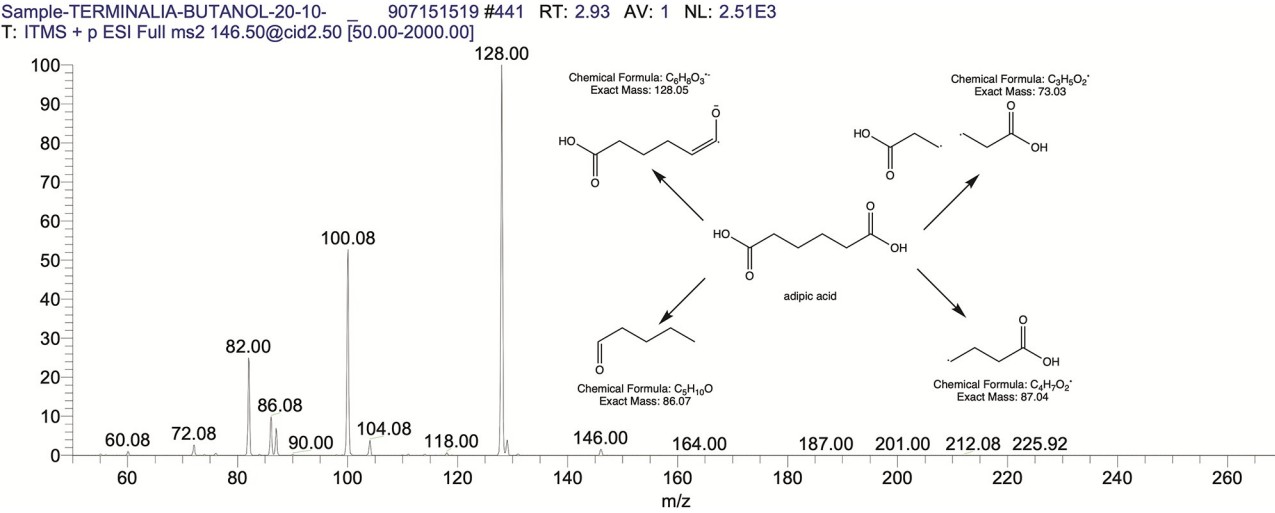

**Fig 17. The mass spectrum of adipic acid and its fragmentation pattern.**

protecting them against the TPA-induced carcinogenesis [48]. The scopoletin a phenolic coumarin, inhibited the growth of breast and colon cancer cell lines [49, 50].

The high antioxidant activity of high molecular weight tannins was attributed to the number of phenolic–OH groups. The literature review revealed that casuarinin a hydrolyzable tannin isolated from *T. arjuna* was effective against lung cancer [51].

## 5 Conclusion

The present study concluded that BUAE secured from methanol extract of aerial parts of *Terminalia neotaliala* showed significant antioxidant activity determined by the DPPH, ABTS,

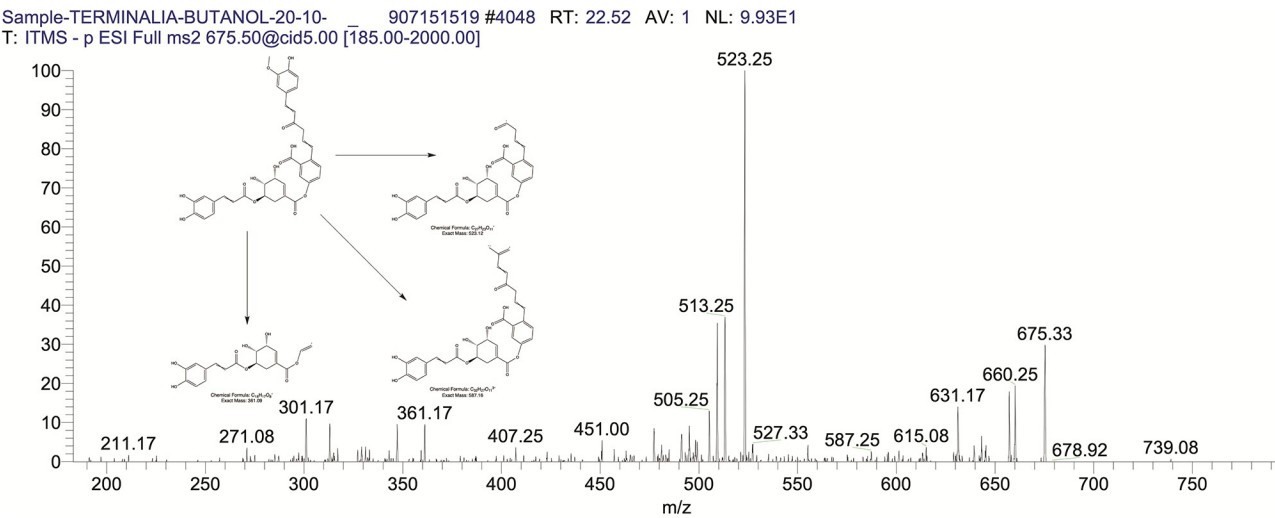

**Fig 18. The mass spectrum of feruloyl-O-p coumaroyl-O caffeoyl shikimic acid and its fragmentation pattern.**

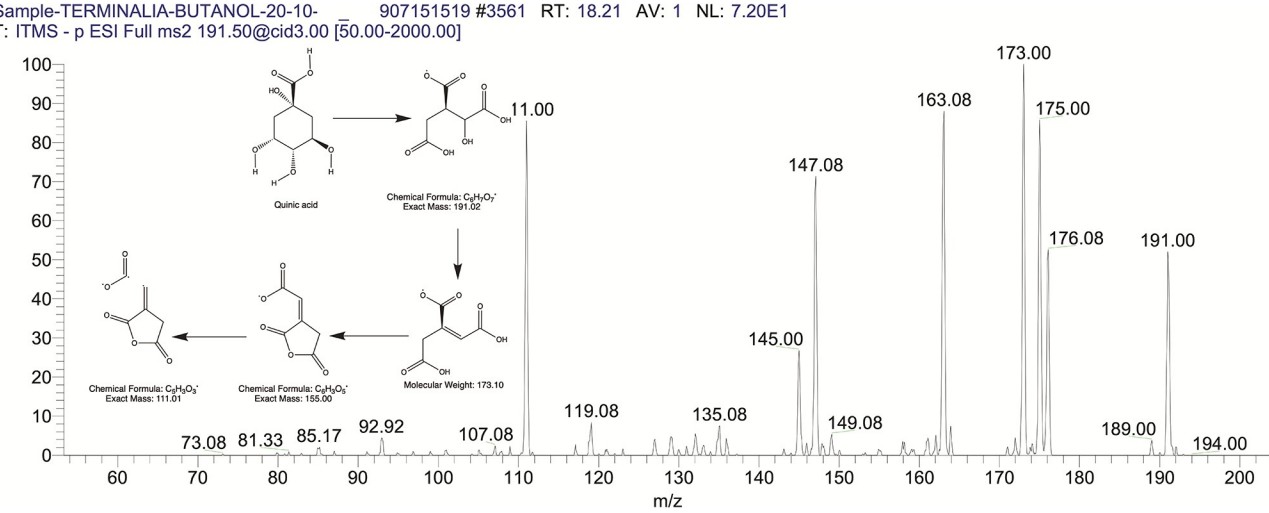

**Fig 19. The mass spectrum of quinic acid and its fragmentation pattern.**

FRAP, and CUPRAC assay. The BUAE also confirmed the highest TTC, TPC, and TFC which were responsible for pharmacological activities validating its ethnobotanical and medicinal importance. LCMS metabolomic profiling of BUAE confirmed the presence of many hydrolyzable tannins in addition to other biologically active phytoconstituents. The strong antioxidant and enzyme inhibitory activity also concluded that the aerial part of *T. neotaliala* can be a promising source of new drug molecules for the treatment and mitigation of different inflammatory conditions, metabolic disorders, and cancer. The BUAE can be further exploited for the discovery of novel pharmaceutical products through different isolation techniques i.e., normal, and reversed-phase column chromatography, preparative TLC, and preparative HPLC.

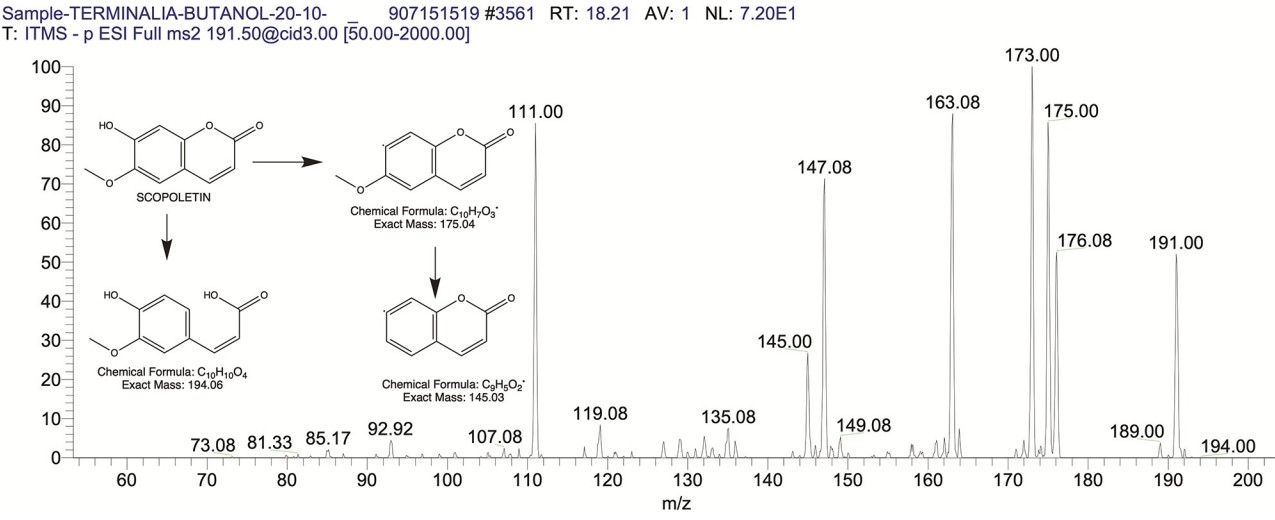

**Fig 20. The mass spectrum of scopoletin and its fragmentation pattern.**

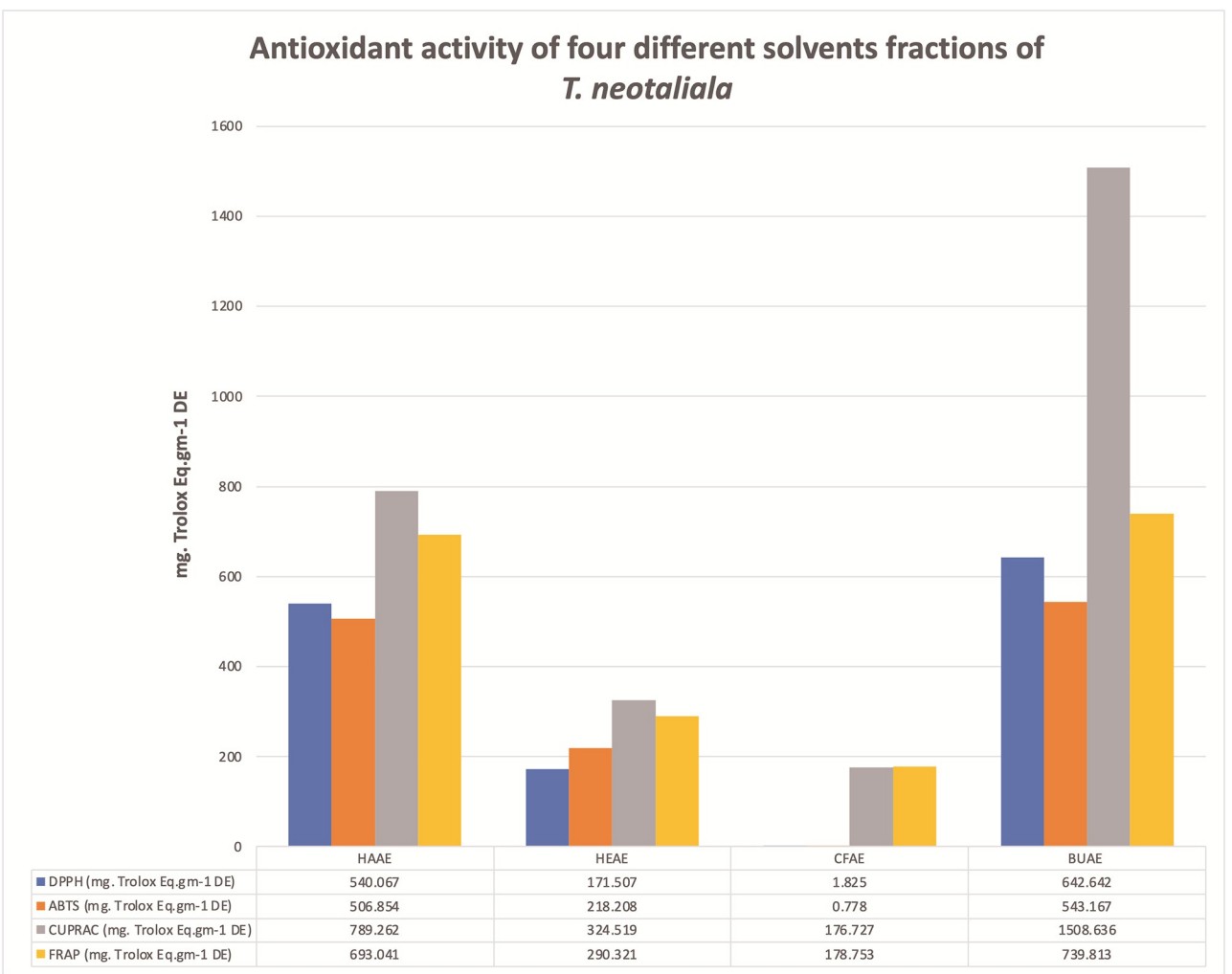

**Fig 21. Antioxidant activity of four sample extracts of *T. neotaliala* determined by DPPH, ABTS, CUPRAC, and FRAP assays.** The antioxidant results were statistically analyzed by one-way ANOVA using SPSS software, that the antioxidant content of each sample extract was significantly different from the antioxidant content of others (p-value < 0.05) which indicated that each solvent had a different capacity to extract phytoconstituents depending on its polarity and chemical structure.

## Author Contributions

**Conceptualization:** Muhammad Nadeem Shahzad.

**Data curation:** Muhammad Nadeem Shahzad, Muhammad Imran Tousif, Imtiaz Ahmad.

**Formal analysis:** Muhammad Nadeem Shahzad, Muhammad Imran Tousif, Imtiaz Ahmad, Huma Rao, Bilal Ahmad.

**Investigation:** Muhammad Nadeem Shahzad.

**Methodology:** Muhammad Nadeem Shahzad.

**Project administration:** Saeed Ahmad.

**Software:** Muhammad Nadeem Shahzad, Imtiaz Ahmad, Huma Rao.

**Supervision:** Saeed Ahmad.

**Validation:** Imtiaz Ahmad.

**Visualization:** Muhammad Nadeem Shahzad, Bilal Ahmad, Abdul Basit.

**Writing – original draft:** Muhammad Nadeem Shahzad, Huma Rao.

**Writing – review & editing:** Abdul Basit.

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
