## [Decision Letter · Decision Letter 0]

26 Nov 2021

PONE-D-21-31614Profiling of phytochemicals from aerial parts of Terminalia neotaliala using LC-ESI-MS and determination of antioxidant and enzyme inhibition activitiesPLOS ONE

Dear Dr. Shahzad,

Thank you for submitting your manuscript to PLOS ONE. After careful consideration, we feel that it has merit but does not fully meet PLOS ONE’s publication criteria as it currently stands. Therefore, we invite you to submit a revised version of the manuscript that addresses the points raised during the review process.

ACADEMIC EDITOR: As appended below, the reviewers have raised major concerns/critiques and suggested further justification/work to consolidate the findings. Do go through the comments and amend the MS accordingly.

We look forward to receiving your revised manuscript.

Kind regards,

A. M. Abd El-Aty

Academic Editor

PLOS ONE

2. PLOS requires an ORCID iD for the corresponding author in Editorial Manager on papers submitted after December 6th, 2016. Please ensure that you have an ORCID iD and that it is validated in Editorial Manager. To do this, go to ‘Update my Information’ (in the upper left-hand corner of the main menu), and click on the Fetch/Validate link next to the ORCID field. This will take you to the ORCID site and allow you to create a new iD or authenticate a pre-existing iD in Editorial Manager. Please see the following video for instructions on linking an ORCID iD to your Editorial Manager account: https://www.youtube.com/watch?v=_xcclfuvtxQ. 

3. Please ensure that you refer to Figure 2 to 22 in your text as, if accepted, production will need this reference to link the reader to the figure

4. Please upload a copy of Figures 17 to 22 to which you refer in your text on page 20. If the figure is no longer to be included as part of the submission please remove all reference to it within the text.

Reviewers' comments:

Reviewer's Responses to Questions

**Comments to the Author**

1. Is the manuscript technically sound, and do the data support the conclusions?

Reviewer #1: Partly

Reviewer #2: Yes

Reviewer #3: Yes

2. Has the statistical analysis been performed appropriately and rigorously? 

Reviewer #1: Yes

Reviewer #2: Yes

Reviewer #3: Yes

3. Have the authors made all data underlying the findings in their manuscript fully available?

Reviewer #1: No

Reviewer #2: Yes

Reviewer #3: Yes

4. Is the manuscript presented in an intelligible fashion and written in standard English?

Reviewer #1: No

Reviewer #2: Yes

Reviewer #3: Yes

5. Review Comments to the Author

Reviewer #1: The authors have made huge effort to address previously raised issues and improve the quality of their manuscript for re-submission. However, I still have doubt on the structural identification process. As noted before, the authors stated that they identified the polyphenols from butanol extract by comparing with previous literature and using LC-MS/MS. I suggested the authors to explicitly describe the fragmentation pattern of each molecule, and also support their identification process and compound elution order using available standards. However, the authors did not conduct or incorporate any of these in the resubmitted manuscript. They sated that they cannot conduct additional experiment/analysis for unknown reason. Therefore, I still consider that these issues are critical and must be fully addressed for the manuscript to be considered for publication.

Reviewer #2: The manuscript PONE-D-21-31614 was revised for English language and the quality of the presentation and description of the data have been notably improved. The quality of figures 2-22 must be improved and then the work can be accepted and be published in Plos One

Reviewer #3: This study showed the Phytochemical profiling of Terminalia neotaliala by LC-ESI-MS: Antioxidant and enzyme inhibition activity. This paper showed Strong antioxidant activity and presence of high molecular weight hydrolysable tannins by LC-ESI-MS suggest medicinal importance of T. neotaliala. Therefore, I would like to suggest to be accepted this manuscript after the improvement as a whole.

1. Line 54, Total phenolic content (TPC) → total phenolic content (TPC)

2. Line 56, Total tannin content (TTC) → TTC

3. Line 116, 143, 160,….etc. in manusctript, figure , table, Terminalia neotaliala → T. neotaliala. You did not revise at all. You should write full name (Terminalia neotaliala) when you firstly mention the academic name in manuscript, figure, table and then should write as abbreviation (T. neotaliala)

4. Table 4, you should check and revise the subscript in formula.

6. PLOS authors have the option to publish the peer review history of their article (what does this mean?). If published, this will include your full peer review and any attached files.

Reviewer #1: No

Reviewer #2: No

Reviewer #3: No

---

## [Author Response · Author response to Decision Letter 0]

4 Jan 2022

Response to reviewer 1: as per suggestion by the reviewer 1, the manuscript was improved by incorporation of each molecular fragmentation pattern after extensively reviewing the available literature. The quality of each molecular ionogram was also improved. The molecule (132-Hydroxypheophorbide-α-methyl ester) incorporated in table 4, serial # 2 with retention time 5.15 and molecular weight 622 gm/mol belongs to chloroform fraction of T. neotaliala which was mistakenly incorporated in butanol fraction (subjected) LC-ESI-MS2 molecular table, due to this author were questioned about the order of elution like higher molecular weight eluting before the low molecular weight compounds. Now the mistakenly incorporated misfit compound has been removed from the list of identified compounds. Kindly find the ion-gram of mentioned molecular compound which clearly indicates the chloroform fraction in its description with further details. The authors could not go for further analysis because the LC-ESI-MS2 facility was availed from other institute who unfortunately did not have all of the identified standards therefore it’s not feasible for authors to buy each of the standard compounds and run them again due to financial limitations. Therefore, authors have extensively reviewed the previous literature and included the fragmentation patter of each identified compound for supporting the identification process to satisfy the reviewer 1.

Response to Reviewer 2. 

The quality of English and scientific language has been improved. Similarly, quality of figures 2-21 have been improved at their best quality for possible publication of our manuscript in Plos One.

response to Reviewer 3. 

Thanks for the positive comments of the reviewer. The suggested corrections have been made in the revised manuscript.

---

## [Decision Letter · Decision Letter 1]

17 Jan 2022

PONE-D-21-31614R1Profiling of phytochemicals from aerial parts of Terminalia neotaliala using LC-ESI-MS and determination of antioxidant and enzyme inhibition activitiesPLOS ONE

Dear Dr. Shahzad,

Thank you for submitting your manuscript to PLOS ONE. After careful consideration, we feel that it has merit but does not fully meet PLOS ONE’s publication criteria as it currently stands. Therefore, we invite you to submit a revised version of the manuscript that addresses the points raised during the review process.

ACADEMIC EDITOR: Still reviewer is raising some comments over the revised form of the MS. Do go through the comments and amend the MS, accordingly. Afterwards, proofread the text for grammar and syntax errors, if any.

We look forward to receiving your revised manuscript.

Kind regards,

A. M. Abd El-Aty

Academic Editor

PLOS ONE

Journal Requirements:

Reviewers' comments:

Reviewer's Responses to Questions

**Comments to the Author**

1. If the authors have adequately addressed your comments raised in a previous round of review and you feel that this manuscript is now acceptable for publication, you may indicate that here to bypass the “Comments to the Author” section, enter your conflict of interest statement in the “Confidential to Editor” section, and submit your "Accept" recommendation.

Reviewer #1: (No Response)

2. Is the manuscript technically sound, and do the data support the conclusions?

Reviewer #1: Partly

3. Has the statistical analysis been performed appropriately and rigorously? 

Reviewer #1: No

4. Have the authors made all data underlying the findings in their manuscript fully available?

Reviewer #1: Yes

5. Is the manuscript presented in an intelligible fashion and written in standard English?

Reviewer #1: No

6. Review Comments to the Author

Reviewer #1: The authors have addressed most of the issues and improved the quality of the manuscript. Once gain, I suggest the authors consider the following minor comments.

-It is preferable to avoid the use of abbreviations while listing keywords.

-Please reflect the statistical analysis data in figure 1 (i.e show if there is any significant variation in each of the antioxidant activities according to extract types).

-Figures are duplicated and hence, need revision.

-The authors have already indicated that the results are reported as mean ± SD (lines 306-307). Thence, it is better to avoid the unnecessary repetition of "SD" while mentioning numerical values in the result and discussion sections (for instance: 234.79 ± 0.12 mg. GAEq.g-1 DE±SD can be rewritten as 234.79 ± 0.12 mg GAEq g-1 DE (line 324), and so forth).

-In the reference section, there is a huge inconsistency in reference listing. Authors are advised to follow the journal's guideline.

-Last but not least, the manuscript still needs a thorough edition and revision towards its language use (preferably by a native speaker).

7. PLOS authors have the option to publish the peer review history of their article (what does this mean?). If published, this will include your full peer review and any attached files.

Reviewer #1: No

---

## [Author Response · Author response to Decision Letter 1]

22 Feb 2022

Reviewer #1: (suggestions)

1. It is preferable to avoid the use of abbreviations while listing keywords.

Response

The suggestion has complied while listing keywords 

2. please reflect the statistical analysis data in figure 1 (i.e. show if there is any significant variation in each of the antioxidant activities according to extract types).

Response

A statement has been incorporated after statistical analysis 

3. Figures are duplicated and hence, need revision.

Response

Duplicate figures have been removed 

4. The authors have already indicated that the results are reported as mean ± SD (lines 306-307). Thence, it is better to avoid the unnecessary repetition of "SD" while mentioning numerical values in the result and discussion sections (for instance: 234.79 ± 0.12 mg. GAEq.g-1 DE±SD can be rewritten as 234.79 ± 0.12 mg GAEq g-1 DE (line 324), and so forth).

Response 

The said suggestion has complied throughout the manuscript

5. In the reference section, there is a huge inconsistency in the reference listing. Authors are advised to follow the journal's guidelines.

Response

The suggestion has complied in the bibliography 

6. Last but not least, the manuscript still needs a thorough edition and revision towards its language use (preferably by a native speaker).

Response 

The manuscript has been completed edited and revised towards its language

---

## [Decision Letter · Decision Letter 2]

28 Feb 2022

PONE-D-21-31614R2Profiling of phytochemicals from aerial parts of Terminalia neotaliala using LC-ESI-MS2 and determination of antioxidant and enzyme inhibition activitiesPLOS ONE

Dear Dr. Shahzad,

Thank you for submitting your manuscript to PLOS ONE. After careful consideration, we feel that it has merit but does not fully meet PLOS ONE’s publication criteria as it currently stands. Therefore, we invite you to submit a revised version of the manuscript that addresses the points raised during the review process.

ACADEMIC EDITOR: As raised by the reviewer, the MS needs language editing  for grammar and syntax errors. Check also Fig. 1 as stated in the comments.

We look forward to receiving your revised manuscript.

Kind regards,

A. M. Abd El-Aty

Academic Editor

PLOS ONE

Journal Requirements:

Reviewers' comments:

Reviewer's Responses to Questions

**Comments to the Author**

1. If the authors have adequately addressed your comments raised in a previous round of review and you feel that this manuscript is now acceptable for publication, you may indicate that here to bypass the “Comments to the Author” section, enter your conflict of interest statement in the “Confidential to Editor” section, and submit your "Accept" recommendation.

Reviewer #1: All comments have been addressed

2. Is the manuscript technically sound, and do the data support the conclusions?

Reviewer #1: Partly

3. Has the statistical analysis been performed appropriately and rigorously? 

Reviewer #1: No

4. Have the authors made all data underlying the findings in their manuscript fully available?

Reviewer #1: Yes

5. Is the manuscript presented in an intelligible fashion and written in standard English?

Reviewer #1: No

6. Review Comments to the Author

Reviewer #1: The authors have addressed most of the issues raised during the two rounds of revisions. The manuscript could be accepted for publication provided that an extensive revision and edition is made towards its language. Besides, figure 1 still needs to show statistical analysis results.

7. PLOS authors have the option to publish the peer review history of their article (what does this mean?). If published, this will include your full peer review and any attached files.

Reviewer #1: No

---

## [Author Response · Author response to Decision Letter 2]

9 Mar 2022

Reviewer #1: Response to comments

1. The manuscript has been completely revised and edited, toward the English language. 

2. The statistical result of figure 1 has been sufficiently described.

---

## [Decision Letter · Decision Letter 3]

15 Mar 2022

Profiling of phytochemicals from aerial parts of Terminalia neotaliala using LC-ESI-MS2 and determination of antioxidant and enzyme inhibition activities

PONE-D-21-31614R3

Dear Dr. Shahzad,

We’re pleased to inform you that your manuscript has been judged scientifically suitable for publication and will be formally accepted for publication once it meets all outstanding technical requirements.

Kind regards,

A. M. Abd El-Aty

Academic Editor

PLOS ONE

Additional Editor Comments (optional):

Reviewers' comments:

Reviewer's Responses to Questions

**Comments to the Author**

1. If the authors have adequately addressed your comments raised in a previous round of review and you feel that this manuscript is now acceptable for publication, you may indicate that here to bypass the “Comments to the Author” section, enter your conflict of interest statement in the “Confidential to Editor” section, and submit your "Accept" recommendation.

Reviewer #1: All comments have been addressed

2. Is the manuscript technically sound, and do the data support the conclusions?

Reviewer #1: Yes

3. Has the statistical analysis been performed appropriately and rigorously? 

Reviewer #1: Yes

4. Have the authors made all data underlying the findings in their manuscript fully available?

Reviewer #1: Yes

5. Is the manuscript presented in an intelligible fashion and written in standard English?

Reviewer #1: Yes

6. Review Comments to the Author

Reviewer #1: The authors have addressed the issues raised during the three rounds of revisions. Now, the manuscript can be considered for publication.

7. PLOS authors have the option to publish the peer review history of their article (what does this mean?). If published, this will include your full peer review and any attached files.

Reviewer #1: No

---

## [Editor Report · Acceptance letter]

18 Mar 2022

PONE-D-21-31614R3 

Profiling of phytochemicals from aerial parts of *Terminalia neotaliala* using LC-ESI-MS^2^ and determination of antioxidant and enzyme inhibition activities 

Dear Dr. Shahzad:

I'm pleased to inform you that your manuscript has been deemed suitable for publication in PLOS ONE. Congratulations! Your manuscript is now with our production department. 

Kind regards, 

on behalf of

Prof. A. M. Abd El-Aty 

Academic Editor

PLOS ONE